# Derived Polymorphic Amplified Cleaved Sequence (dPACS): A Novel PCR-RFLP Procedure for Detecting Known Single Nucleotide and Deletion–Insertion Polymorphisms

**DOI:** 10.3390/ijms20133193

**Published:** 2019-06-29

**Authors:** Shiv Shankhar Kaundun, Elisabetta Marchegiani, Sarah-Jane Hutchings, Ken Baker

**Affiliations:** 1Herbicide Bioscience, Syngenta Ltd., Jealott’s Hill International Research Centre, RG42 6EY Bracknell, UK; 2General Bioinformatics, Jealott’s Hill International Research Centre, RG42 6EY Bracknell, UK

**Keywords:** dPACS, PCR-RFLP, SNP, genotyping, herbicide resistance, acetyl-CoA carboxylase (ACCase), photosystem II, EPSPS

## Abstract

Most methods developed for detecting known single nucleotide polymorphisms (SNP) and deletion–insertion polymorphisms (DIP) are dependent on sequence conservation around the SNP/DIP and are therefore not suitable for application to heterogeneous organisms. Here we describe a novel, versatile and simple PCR-RFLP procedure baptised ‘derived Polymorphic Amplified Cleaved Sequence’ (dPACS) for genotyping individual samples. The notable advantage of the method is that it employs a pair of primers that cover the entire fragment to be amplified except for one or few diagnostic bases around the SNP/DIP being investigated. As such, it provides greater opportunities to introduce mismatches in one or both of the 35–55 bp primers for creating a restriction site that unambiguously differentiates wild from mutant sequences following PCR-RFLP and horizontal MetaPhor^TM^ gel electrophoresis. Selection of effective restriction enzymes and primers is aided by the newly developed dPACS 1.0 software. The highly transferable dPACS procedure is exemplified here with the positive detection (in up to 24 grass and broadleaf species tested) of wild type proline106 of 5-enolpyruvylshikimate-3-phosphate synthase and its serine, threonine and alanine variants that confer resistance to glyphosate, and serine264 and isoleucine2041 which are key target-site determinants for weed sensitivities to some photosystem II and acetyl-CoA carboxylase inhibiting herbicides, respectively.

## 1. Introduction

Single nucleotide polymorphisms (SNPs) are the most common type of variation in the genome [1]. In humans, there are as many as 10 million SNPs or around one in every 1000 nucleotides [2]. When they occur in exons and cause a change in protein structure and function, they may have profound phenotypic effects on the organism [3]. Several such SNPs are documented including an adenine to thymine change in the β-globin gene, resulting in sickle-cell anaemia [4]. Other similar and well-known human diseases that are due to non-synonymous nucleotide changes in exons comprise of phenylketonuria, β-thalassemia and cystic fibrosis [5,6,7]. Base changes that are located in non-coding regions may still impact on gene splicing and transcription factor binding [8,9]. For instance, a cytosine to thymine transition in intron 8 of bovine *CD46* results in the retention of an additional 48 base pair fragment in the splice variant. The ensuing 16 amino acid enlarged protein is linked to mastitis in Chinese Holstein cows [10]. SNPs that occur in inter-genic regions are generally silent phenotypically but are still useful as markers in genome-wide association and evolutionary and population genetics studies [11,12,13]. While not as frequent as SNPs, deletion–insertion polymorphisms (DIPs or indels) are widely spread in the genome [14]. They are also produced as a result of the increasingly employed CRISPR/CAS9 technology for genome editing [15]. Indels which are multiples of three nucleotides will maintain the open reading frame of genes but result in shorter/longer amino acid strands potentially altering the structure and function of proteins [16]. Indels which are not a multiple of three nucleotides give rise to frameshift mutations, thereby coding for an entirely different set of amino acids or resulting in a premature stop codon [17].

Because of their abundance and importance, a flurry of methods have been developed to detect SNPs and DIPs [18,19]. These can be divided in a number of ways including procedures for identifying unknown and known SNPs/DIPs [20]. The choice of the detection technique is dependent on several factors such as robustness, throughput and costs [21]. In any case, all the methods consist of two main steps: the biochemical reactions for allele discrimination and detection procedures for identifying the products [18]. Most of the assays separate these two processes although they can be carried out in parallel to minimise handling steps and increase assay throughput. This is the case of the TaqMan genotyping assay which is a single-step, closed-tube approach capable of differentiating between homozygous and heterozygous mutant individuals [22,23]. Other PCR-based, sophisticated but also costly, genotyping methods include pyrosequencing, SNaPshot and Next Generation Sequencing (NGS) [24,25,26]. In spite of a large number of cutting-edge assays developed in recent years there is still need for simple procedures that require low technical skills and initial investments. One relatively cheap methodology that was described soon after the invention of polymerase chain reaction is the Allele Specific Amplification (ASA) technique in which one of the primers is designed to allow amplification by DNA polymerase only if the 3′ end of the primer perfectly matches the wild type or variant nucleotide(s) [27]. The positive identification of the wild and/or mutant allele is manifested by the presence of an expected PCR fragment(s) that can be visualised on simple horizontal agarose gel electrophoresis. Targeted DNA amplification via PCR, followed by restriction fragment length polymorphism developed more than 30 years ago, also remains a common and simple genotyping method [28,29]. Two PCR-RFLP methodologies are described, namely, the Cleaved Amplified Polymorphic Sequence (CAPS) and the derived Cleaved Amplified Polymorphic Sequence (dCAPS) assays [30,31]. The CAPS assay takes advantage of the creation or loss of a discriminating restriction site as a consequence of a base change in the DNA sequence being investigated [31]. Polymorphism in the form of different electrophoretic band sizes is revealed following digestion of the PCR fragment with a convenient restriction enzyme. If the SNP does not lead to a loss or gain of a restriction site with a commonly available enzyme, the dCAPS assay can be employed [30]. In this case, one of the two primers is designed to anneal adjacent to the SNP and in which one or a few mismatches with respect to the template DNA are introduced to create a restriction site that differentiates wild from mutant sequences. Selection of primers with one or more mismatches and corresponding restriction enzymes is facilitated by the use of the dCAPS Finder 2.0 freeware [32].

Here we propose a variant of the PCR-RFLP approach named the ‘derived Polymorphic Amplified Cleaved Sequence’ (dPACS) which provides greater possibilities for genotyping SNPs and DIPs compared to the CAPS and dCAPS methodologies. The dPACS assay utilises a primer pair that encompasses the whole DNA region to be amplified except for the SNP/DIP or, as in the case of this study, bar the two bases that characterise the amino acid codon being genotyped. The use of primers located very close to the SNP or codon triplet offers the prospect for introducing forced mutations in either or both forward and reverse primers to allow differentiation between wild and mutant DNA with appropriate restriction enzymes. Discriminant primer and enzyme combinations are generated with the help of a novel custom design dPACS 1.0 software. The methodology and associated advantages are illustrated with the positive identification of nucleotides that code for proline106 (and its serine, threonine and alanine variants), serine264 and isoleucine2041 of 5-enolpyruvylshikimate-3-phosphate synthase (EPSPS), *psb*A and acetyl-CoA carboxylase (ACCase) genes respectively, in a range of weed species. Alteration of these key amino acid residues hampers the proper binding of some of the commonly employed EPSPS, photosystem II and ACCase-inhibiting herbicides allowing weeds to survive, causing significant yield loss in major cropping systems worldwide [33].

## 2. Results

### 2.1. S264-psbA dPACS Assay

Polymerase chain reaction amplified an expected 426 bp fragment for all the 48 *Amaranthus retroflexus* plants from the six different populations. Nucleotide sequence analysis of the PCR fragment encompassing codons 165 to 307 showed on average 99.9% homology with published *psb*A sequences (e.g., GenBank accession DQ887375.1) supporting the identity of the gene amplified here. A single nucleotide polymorphism was detected among the 48 individuals, namely the targeted adenine to guanine transition at the first base of codon 264, resulting in the serine to glycine mutation in the D1 protein of photosystem II. The resistance-causing nucleotide change was detected in all eight plants from populations ArD1, ArD4 and ArD6 whilst all individuals from populations ArD2, ArD3 and ArD5 contained the wild type nucleotide at this position (Table 1). No single plant contained both wild and mutant alleles which was consistent with the maternal inheritance of chloroplastic genes (*psb*A) in most angiosperms [34].

Wild (264-AGT) and mutant (264-GGT) nucleotide sequences 113 bp long, including 55 bp on each side of the targeted codon, were entered in the dPACS 1.0 program in view of identifying discriminating primer/enzyme combinations that will positively detect the herbicide-sensitive serine amino acid residue at *psb*A codon position 264. The program generated one, four and 20 discriminating primer/enzyme options with zero, one and two forced mutations respectively, on either the forward, reverse or both primers. The chosen enzyme was *Ple*I (recognition site GAGTCN4) which required one mismatch each on the forward and reverse dPACS primers to create a restriction site for plants that contain the serine (AGT codon) 264 allele (Figure 1).

The mismatches consisted of a thymine to guanine and adenine to guanine change on the 3′ end of the forward primer and penultimate base at the 3′ end on the reverse primer, respectively. Using a 40 bp forward primer that ended on the last base of the 263 codon and a 41 bp reverse primer located at the 3rd base of the 264 codon, PCR generated an expected 83 bp dPACS fragment for all plants. Upon restriction with *Ple*I and electrophoresis on 4% MetaPhor^TM^ gel, plants bearing the wild type serine allele generated two smaller fragments of 45 bp and 38 bp that could be easily resolved from the mutant 83 bp undigested PCR fragment (Figure 2).

Analysis of the 48 plants from the six *A. retroflexus* populations completely agreed with the results from the Sanger sequencing analysis (Table 1). The dPACS primers based on *A. retroflexus* sequences successfully amplified a corresponding PCR fragment from all 12 diverse grass and 12 broadleaf species. Upon restriction with *Ple*I, all 192 (24 species × 8 individuals) monocotyledonous and dicotyledonous plants displayed two smaller digested bands of 38 and 45 bp. This was expected as all samples tested in this initial transferability study consisted of wild type herbicide-sensitive individuals. On the other hand, all the ametryn-resistant *Chenopodium album* and *Solanum nigrum* individuals from the three different populations showed an undigested band of 83 bp indicating a mutant allele at *psb*A codon position 264 (Table 2).

### 2.2. I2041-ACCase dPACS Assay

In view of sequencing the ACCase gene around codon 2041, a primer pair was employed to encompass the 3rd base of codon 1954 and the 2nd base of position codon 2110. Sequencing of the resulting 468 bp fragment showed 98% homology with ACCase GenBank sequence AY710293. Nineteen base changes were detected among the 48 *Lolium multiflorum* plants. These include the thymine to adenine transversion at the second base of codon 2041 resulting in the ACCase herbicide resistance causing isoleucine (ATT) to asparagine (AAT) mutation. The I2041N mutation was detected in a few individuals from three (LmD1, LmD5 and LmD6) out of the six populations used for developing the dPACS-2041 assay (Table 1). Of relevance for primer and assay design is a synonymous thymine to cytosine transition at the third base of the 2041 codon (Figure 1). Submission of relevant wild and mutant nucleotide sequences in the dPACS program did not reveal any restriction enzyme that will positively identify the isoleucine residue at codon position 2041 (taking into consideration the silent mutation on the 3rd base of codon 2041) with zero mismatches on the primers. With one forced mutation allowed, two discriminant enzymes were identified, namely, *Hinf*III which is not commercially available, and the chosen *Eco*RI which necessitates fixing the 3rd variable base of the 2041 codon triplet to a thymine with the reverse primer. PCR carried out with 35 bp forward and 55 reverse dPACS primers bordering the AT dinucleotides characteristic of isoleucine at ACCase codon position 2041 amplified a corresponding 92 bp fragment for all the *L. multiflorum* plants initially analysed. Upon restriction with *Eco*RI and electrophoresis on 4% MetaPhor^TM^ gel, wild type plants showed two smaller fragments of 58 and 34 bps (Figure 3).

Homozygous mutant plants were characterised by the undigested 92 base pair PCR fragment whilst heterozygous individuals displayed one copy each of the wild type restricted (58, 34 bp) and the unrestricted 92 bp fragments. The data generated with the 2041-ACCase dPACS assay totally matched those from DNA sequencing (Table 1). PCR with 96 (12 × 8 individuals) additional ACCase herbicide-sensitive plants belonging to 12 diverse grass weed species also amplified a corresponding 92 bp band which upon restriction with *Eco*RI produced the predicted wild type 58 bp and 34 bp restricted fragments, thereby exhibiting the wide transferability of the 2041-dPACS assay. Analysis of six each of *Setaria viridis* and *Eleusine indica* populations that are resistant to the herbicide haloxyfop-p-methyl identified mutant 2041-ACCase plants in one (SvR2) and three (EiR2, EiR3 and EiR6) populations respectively. The corresponding genotypic frequencies at 2041-ACCase codon are provided in Table 2.

### 2.3. P106-EPSPS dPACS Assay

The targeted 106-EPSPS codon is located 39–41 bp from the 3′ end of the 245 bp exon 2 in *E. indica*. To capture the nucleotide sequences around codon 106, a pair of primers was employed that extends from the 5′ end of exon 2 to the 3′ end of exon 3. PCR generated a fragment of 498 or 492 bases among the different plants. Analysis of the nucleotide sequences of exon 2 and 3 showed 99% homology with published *E. indica* EPSPS gene (e.g., AY157642). The nucleotides were particularly conserved among the 48 *E. indica* plants analysed except for the first base of codon 106 and a short indel of six base pairs in intron 2. Three different codon triplets were observed at position 106, namely those that code for wild type proline (CCA) and mutant serine (TCA) and alanine (GCA) amino acid residues (Table 1). All plants from populations EiD4, EiD1/EiD2 and EiD3 consisted of homozygous

PP106-EPSPS, SS106-EPSPS and AA106-EPSPS individuals, respectively. Populations EiD5 and EiD6 contained a mixture of wild and mutant plants bearing the proline and serine106 alleles at different genotypic frequencies (Table 1). As different allelic variants occur at EPSPS-codon 106, an assay was attempted to allow characterisation of the wild type proline allele and the commonly encountered mutant alleles using a single PCR product but different restriction enzymes (Figure 1). The dPACS 1.0 program identified the non-interfering restriction enzymes *Xcm*I, *Xmn*I, *Rsa*I and *Cac*8I for positively detecting the proline, serine, threonine and alanine amino acid residues at EPSPS codon 106, respectively. Since a restriction site is naturally created for *Cac*8I as a consequence of the P106A-EPSPS mutation, no additional mismatch was required in the primers. On the other hand, two and three forced base changes are necessary on the forward and reverse primers respectively, to create a restriction site for *Xmn*I and *Xcm*I for detecting the serine and proline alleles. The identification of the threonine allele with *Rsa*I necessitates a single base change at the 3′end of the forward primers which is a common requirement for identifying the serine allele with *Xmn*I. Using 40 bps forward and reverse primers covering the whole region around EPSPS codon 106 except for the characteristic dinucleotides at the first and second base of the codon, PCR amplified an 82 bp fragment for all the 48 *E. indica* plants that were previously sequenced. Restriction digestion with the four different enzymes in separate reactions produced smaller digested fragments only when the respective alleles were present in the plant samples (Table 1). Examples of clearly separated wild and homozygous and heterozygous mutant profiles and corresponding fragment sizes with the different enzymes are shown on Figure 4 and Table 3.

Low levels of undigested PCR fragments (which sometime occur with PCR-RFLP approaches) in lanes 2, 7 and 9 of Figure 4 should not be misinterpreted as heterozygous individuals given that they are of much lower intensity that the corresponding restricted bands. Importantly, the data generated with the EPSPS-dPACS assays completely agreed with those obtained with Sanger sequencing. The primers based on *E. indica* nucleotide sequences successfully amplified a strong 82 bp band in 96 herbicide-sensitive plants from 12 other grass but not to an intensity adequate for downstream RFLP analysis for the 12 broad-leaf weed species. A corresponding PCR fragment was generated from all broadleaf weeds but *Polygonum convolvulus* using a pair of equivalent primers based on *Amaranthus tuberculatus*. Restriction analysis with the enzyme *Xcm*I of the targeted EPSPS fragment from herbicide-sensitive plants belonging to 23 diverse species produced two smaller fragments typical of the wild type homozygous proline amino acid residue. The wild-type proline dPACS profile was also observed for all the six herbicide-resistant *Amaranthus palmeri* populations from the USA (Table 2). In contrast, all the glyphosate-resistant *Digitaria insularis* plants from Brazil were heterozygous for wild type proline and either a mutant serine or threonine alleles (Table 2 and Figure 5).

## 3. Discussion

The analysis of DNA sequence variation is performed in a wide range of disciplines including biomedical sciences, marker-assisted breeding and population genetics studies [35,36,37]. It is therefore not surprising that new genotyping technologies are constantly being developed since the first description, more than 40 years ago, of the laborious, time consuming and costly radiolabelled Southern blotting procedure [38]. Current genotyping strategies are generally PCR-based, using a combination of a few tactics only for allele discrimination and signal detection [18,39]. Importantly, SNP/DIP detection methods have dramatically improved over time, especially with regard to sample throughput and accuracy. It is now possible to analyse hundreds of thousands of SNPs in a few samples with DNA chip-based methodologies [40]. When a large number of individuals require genotyping for a few SNPs/DIPs, the probe-based Taqman and the more recent Kompetitive Allele Specific PCR (KASP) technology are preferred [41,42]. Pyrosequencing and amplicon-seq are other useful tools for detecting any new polymorphisms at and around the targeted nucleotide(s) and for ascertaining that the correct locus is being analysed [43,44]. High-throughput sophisticated systems such as the matrix-assisted laser desorption ionization-time of flight mass spectrometry (*MALDI*-*TOF* MS) for allele detection and NGS are, however, often well beyond the budget of most laboratories because of instrumentation costs and/or bioinformatics requirements [45,46]. It explains why simple procedures such as CAPS and dCAPS remain the most commonly employed techniques for the low to medium throughput analysis of SNPs and DIPs [28,30]. This is attested by numerous recent studies that have used PCR-RFLP methodologies both in research and as a routine molecular diagnostic tool. For instance, CAPS assays were successfully developed as potential biomarkers for oral cancer [47], for contributing to a paradigm shift and clearly establishing male heterogamy in two henophidian snakes [48], for molecular sexing in the endangered Galapagos petrel [49] and for ascertaining the identity of the amoebiasis-causing *Entamoeba histolytica* [50]. The same approach was employed for the rapid diagnosis of neonatal sepsis in India [51] and to determine the magnitude of drug resistance in *Mycobacterium tuberculosis* in Central Punjab, Pakistan [52]. Other recent examples of the use of dCAPS markers abound in phylogeny [53], map-based cloning [54], gene flow [55,56] and population genetics studies [57].

The novel dPACS assay developed in this study will benefit the many specialities that use PCR-RFLP and other procedures for analysing SNPs and DIPs. It is a relatively inexpensive assay requiring basic equipment and consumables commonly available in all molecular biology laboratories. The dPACS technique generates co-dominant markers allowing clear differentiation between homozygous and heterozygous individuals. It proceeds in three steps, namely, polymerase chain reaction encompassing the SNP/DIP region followed by restriction digestion and gel electrophoresis. The dPACS primers should be at least 35 bp long to permit clear differentiation between restricted and non-restricted PCR fragments, but not above 55 bp to avoid unnecessary costs associated with extra purification steps post chemical synthesis by primer providers. As the digestion products are relatively small and the difference between the restricted fragments is as low as a few base pairs in many cases, MetaPhor^TM^ gel with better resolving power is required instead of the classical agarose gel used in CAPS and dCAPS methodologies [28,30].

The primers and enzymes are selected by the newly developed dPACS 1.0 program that can identify forced mutations on the forward, reverse or both primers simultaneously for differentiating wild from mutant sequences. On the other hand, the dCAPS freeware will only select discriminating endonucleases based on nucleotide mismatches on either the forward or reverse primers due to the intrinsic nature of the methodology [32]. The dPACS 1.0 program can handle stretches of targeted DNA of over 100 bp (restricted to 60 nucleotides for dCAPS Finder 2.0) and importantly indels and wild/mutant sequences that differ by more than one nucleotide. The ability to detect DIPs is particularly important in this era of CRISPR/CAS9 gene-editing technology which often creates small indels [58,59]. It can be queried for the desired number of nucleotide mismatches on the primers whilst this is limited to three for the dCAPS Finder 2.0 freeware [32]. Furthermore, the dPACS 1.0 program will detect non-specific restriction sites within the amplified DNA fragment for the chosen enzyme. Subsequent elimination of additional recognition sites with forced mutations on the primers is important for the straightforward interpretation of gel electrophoresis data. In contrast, disruption of supplementary recognition sites is rarely possible with the dCAPS approach as the primers only cover a small portion of the amplified PCR fragment [60,61]. For instance, *Mse*I was not considered for the analysis of a cytosine to thymine transition in *PHYA* of Arabidopsis because the recognition site TTAA is frequently encountered in the gene [30]. This problem does not arise when employing the dPACS procedure as all undesirable nucleotides can be modified with the primers.

The dPACS approach has several other advantages over existing PCR-RFLP procedures, mainly as a result of both forward and reverse primers being adjacent to the targeted SNP and DIP. Inclusion of mismatches on both primers offer greater opportunities for SNP/DIP examination compared to the CAPS/dCAPS strategies. With one forced mutation on each primer, the dPACS approach provided 14, 11 and 9 additional primer/enzyme combinations for positively identifying the wild type S264-*psb*A, I2041-ACCase and P106-EPSPS alleles studied here. A further 42, 33 and 56 primer/enzyme options were generated for detecting the corresponding wild type alleles with three forced mutations straddled over the two primers. Several published PCR-RFLP assays could be improved, especially in terms of restriction enzyme cost and reliability, with the additional primer/enzyme possibilities presented by the dPACS method. For example, the *Sph*I-based dCAPS marker linked to smut resistance in corn lines is prone to false negatives of resistance because the enzyme recognition site comprises three nucleotides that are not part of the dCAPS primer and the SNP being investigated [62]. This ambiguity could be easily removed by using the same diagnostic dCAPS forward primer/*Sph*I enzyme combination and by fixing the three extra nucleotides 3′ of the SNP with the reverse dPACS primer. Similarly, a more cost-effective *Eco*RV or *Hind*III-based dPACS assay with one mutation each on the forward and reverse primers could be developed to identify an adenine insertion in *Pun*I causing a loss in pungency in sweet pepper [63]. *Eco*RV and *Hind*III are respectively eight and 20 times less expensive than *Ms*eI used in the corresponding dCAPS method.

The availability of both dPACS primers for creating a discriminating restriction site may also permit the detection of two closely located and meaningful point mutations using a single polymerase chain reaction but two diagnostic restriction enzymes, thereby halving the cost associated with PCR. For instance, it could allow the effective detection of two adenine to guanine transitions at nucleotide positions 2142 and 2143 of the 23S rRNA gene conferring resistance to clarithromycin in *Helicobacter pylori* [64]. A second case is represented by the identification of the D376Q and R377H mutations endowing resistance to acetolactate synthase-inhibiting herbicides [65,66]. Additionally, the generation of a positive assay for each haplotype is desirable especially when several mutant allelic variants exist at a specific nucleotide position [67]. Whilst this can sometimes be achieved with the dCAPS procedure [68], the opportunity is greatly increased with mismatches on one of the dPACS primers for positively identifying the wild type allele and forced mutations on the other dPACS primer for detecting the mutant allele. Another scenario where the dPACS approach will prove critical is for the investigation of mutations at the end of exons or on short exons altogether surrounded by long and highly polymorphic introns, examples of which include the herbicide-resistance causing R128G/M mutations and G210 codon deletion on the protoporphyrinogen oxidase (PPO) gene [69,70]. Both the R128 mutation and 210 PPO codon deletion have proved difficult to genotype to date because they are located on exons that are only 51 and 66 base pairs respectively, among other reasons. Consequently the 3′ end of at least one of the primers was positioned on the highly polymorphic indel-containing introns, thus increasing the risk of PCR failures. By the nature of the dPACS methodology, the 3′ end of both primers will be located on the relatively conserved exons, obliterating the issue arising with primers located on variable DNA regions.

The second major difference between the dPACS and other simple CAPS/dCAPS/ASA assays is that it uses relatively long primers (35–55 bp) to amplify a short (around 80–100 bp) PCR fragment. Targeting a short DNA region is significant when using highly degraded templates from ancient materials and processed food [71,72]. The detrimental effect of poor DNA quality on PCR is further compounded by inhibitors of Taq polymerase in food matrix [73]. For example, only PCR amplicons of less than 100 bp were generated in the analysis of *Colla corii asini* or donkey-hide gelatine (employed in traditional Chinese medicine) for potential mule or bovine adulterants [74]. Similarly, an attempt to authenticate *Pulsatilla chinensis* produced mixed results in terms of PCR amplification because relatively long fragments were aimed at in the ITS2/RFLP barcoding approach [75]. The dPACS assay amplifying shorter DNA fragments will prove more efficient than CAPS/dCAPS methodologies in the certification of highly degraded processed materials including the detection of haram DNA in halal products and disease-prone rat and monkey meats in the food chain [76,77].

The SNPs used in this study to illustrate the dPACS methodology are located on three major herbicide target genes. Amino acid changes at these critical codon positions affect the binding and efficacy of the corresponding herbicides, allowing weeds to survive in agricultural and non-agricultural systems alike [33]. The S264G change in the *psb*A gene, for example, renders the commonly applied triazine herbicides completely ineffective at controlling plants and populations bearing this mutation [78]. The I2041N mutation confers high levels of resistance to the aryloxyphenoxypropionate toxophores but only low to no resistance to two other sub-classes of acetyl-CoA carboxylase-inhibiting herbicides [79]. The P106X mutations generally endow low but biologically significant levels of resistance to glyphosate depending on the rate of herbicide applied, environmental conditions, mutant allele involved, heterozygosity of the individual plants and ploidy of the species [80,81,82,83,84]. Due to ‘extreme parallelism’, resistance evolution caused by the same three mutations tend to be selected in genetically and geographically related and unrelated weed species and populations following similar herbicide application regimes [85,86]. Given the prevalence and economic importance of these three amino acid changes, several molecular assays were developed for their speedy identification in a range of herbicide-resistant weeds worldwide. In this respect, two CAPS assays were designed for the detection of the S264G mutation in *C. album* taking advantage of a loss of the *Mae*I/*Fsp*BI restriction site (CTAG) as a consequence of the adenine to guanine transition on the first base of the 264 codon [87,88]. More recently, a bi-directional allelic specific amplification assay was proposed allowing the positive identification of both wild and mutant haplotypes in the six weed species tested [89]. Whilst simple and proceeding in only two steps, namely, PCR followed by gel electrophoresis, the ASA method is prone to false positive and negative detection of resistance as allele discrimination completely relies on a single nucleotide at the 3′end of the forward and reverse primers [90]. In spite of mismatches at the 3′end of the forward primer and N-1 base with respect to the 3′ end of the reverse primer, PCR amplified a strong band for the S264G assay developed here, thus demonstrating the versatility of dPACS methodology. The assay proved very reliable as the dPACS restriction profiles completely agreed with Sanger sequencing results for all the *A. retroflexus* plants analysed. The S264G-dPACS assay was highly transferable to all 24 other diverse weed species without any additional optimisation. Furthermore, it employs the enzyme *Ple*I which is half the price of *Mae*I/*Fsp*BI used in the published CAPS assays, or *Spe*I that could be utilised in a potential dCAPS strategy with only one mismatch on the forward primer. Analysis of three resistant *S. nigrum* and *C. album* populations identified the mutant G264-*psb*A allele as previously observed in other samples that were not controlled with triazine herbicides [91,92]. The assay developed here will not only be useful for genotyping a wide range of resistant weed species and populations but also for certifying mutant S264G-*psb*A canola seed lots before planting, thereby avoiding mis-application of triazine herbicides on wild type susceptible varieties with devastating agronomic consequences [93].

In the same manner, a bi-directional ASA assay and two *Eco*RI-based CAPS methodologies were established for determining the amino acid status at ACCase codon 2041 [94,95]. The ASA assay designed for use on *A. myosuroides* is prone to false positives and negatives of resistance due to the intrinsic nature of the methodology while the *Lolium* specific CAPS assays were unreliable because the *Eco*RI restriction site is contained in the degenerate 2041 codon triplet (ATT or ATC for wild type isoleucine). Upon realisation of the potential nucleotide variability around the 3rd base of the 2041 codon, a dCAPS assay was developed for the positive detection of the AT dinucleotide that defines the wild type isoleucine amino acid residue at this codon position [60]. The dCAPS approach used consensus primers, with the inclusion of a neutral inosine base as required, to ensure broad applicability to all 39 grass weed species tested. However, a nested PCR approach was needed due to the low yield of the first round of amplification. Use of PCR products as DNA template in subsequent reactions is undesirable because of contamination issues [96]. Additionally, the assay required two mismatches at the N-1 and N-2 nucleotide positions with regard to the 3′end of the forward dCAPS primer making it susceptible to PCR failures in the event of a variable third base of the preceding 2040 ACCase codon. This is the case for some *Echinochloa crus-galli* individuals which are characterised by guanine instead of the commonly encountered adenine nucleotide at the 3rd base of codon 2040 (GenBank reference: HQ395758). Furthermore, the dCAPS technique employed the restriction enzyme *Vsp*I which is 20 times more expensive than *Eco*RI utilised in the dPACS assay. The dPACS assay developed using *L. multiflorum* was directly applicable to all the 12 other diverse grass weed species following a single PCR run followed by restriction analysis. Examination of six each of *S. viridis* and *E. indica* resistant populations showed that 90% of the plants contained a wild type isoleucine allele at codon 2041 suggesting the presence of other target-site and/or non-target-site mechanisms as is commonly observed in weeds that have evolved resistance to ACCase-inhibiting herbicides [79].

In spite of resistance evolution to glyphosate in as many as 38 weeds, the P106X target-site mutations have been identified in only a handful of species with a corresponding DNA-based assay established for *E. indica*, *L. rigidum* and *A. tuberculatus* [80,97,98,99,100]. This is because the P106X EPSPS mutations confer relatively low levels of resistance to glyphosate, and also, the target EPSPS sequence being available for less than half of resistant weed species [101]. The methodologies developed comprised two bi-directional ASAs for detecting the commonly encountered P106S mutation in *E. indica* and a rare P106L mutation in *L. multiflorum* [80,99]. In both cases, the allele specific primers were destabilised at the N-1 bases with respect to the 3′ end to increase the reliability of the assays. A single-step, close tube TaqMan assay was also developed to genotype *E. indica* populations for the P106S mutation [98]. Whilst rapid and co-dominant, the TaqMan assay requires relatively costly equipment and labelled probes. Furthermore, it is sensitive to nucleotide variability around the probe region other that the SNP being investigated [18]. As proline is coded by the degenerate CCN triplet at EPSPS codon position 106, the Taqman assay may not be widely applicable to species and populations that contain a different nucleotide at the third base of the codon. More recently, two *Sau96*I and *Nco*I based dCAPS methods were established for detecting the wild type proline allele in *E. indica* [97] and *A. tuberculatus* [100] respectively. As the critical 106 EPSPS codon is located near the end of exon 2, the reverse primer used for *E. indica* was located on exon 3 with the risk of generating complex restriction profiles due to a variable indel-containing intron 2, as observed in this study. The *Nco*I-based *A. tuberculatus* dCAPS assay specifically designed for interrogating the first nucleotide of the 106 EPSPS codon is susceptible to false negatives of resistance in the event of a variable second base of the codon triplet as previously identified in *L. rigidum* and *E. indica* [99,102]. In any case, there was no attempt to test the wide transferability of the published ASA, dCAPS and TaqMan P106-EPSPS assays given the relatively variable *EPSPS* nucleotide sequences among different weed species [80,97,98,99,100]. The P106X-EPSPS dPACS assay developed here used primers entirely located on exon 2 except for the two bases at the 5′end of the reverse primer, thus avoiding any complications that could arise from a variable intron 2. The assay introduced three mismatches in the reverse primer to positively identify the proline wild type allele and two force mutations in the forward primers to detect the serine, threonine and alanine variants. In spite of at least five mismatches in the primers, a strong PCR fragment was amplified for all *E. indica* plants. The restriction profiles totally agreed with Sanger sequencing results, demonstrating the reliability of the methodology. The *E. indica*-based assay was highly transferable to all other grass weeds but not the 12 other broad-leaf species, owing to the significant *EPSPS* sequence difference between these two plants groups. Nevertheless, a second *A. tuberculatus*-based assay was directly transferable to 11 out of 12 dicot grass weeds showing the flexibility of the dPACS technique. Analysis of six glyphosate resistant *A. palmeri* populations reveal a wild type P106 allele in all forty-eight plants. This is not surprising as earlier studies have shown that glyphosate resistance in this species is overwhelmingly due to a target gene duplication [103,104,105,106]. On the other hand, all the 48 *D. insularis* samples contained either a P106S or P106T mutation at the heterozygous state. The threonine allelic variant was previously found in a *D. insularis* population resistant to glyphosate whilst this is the first description of the serine mutant allele in this species [107]. Given the relatively low and variable impact of the P106X-EPSPS mutations on glyphosate efficacy, resistance due to the mutant serine allele in *D. insularis* cannot be assumed from observations in other weed species. The P106X assays developed in this study will be very useful for creating wild homozygous PP106, heterozygous PS106 and mutant homozygous SS106 EPSPS subpopulations for adequately investigating the impact and dominance of the serine106-EPSPS variant on the efficacy of glyphosate in *D. insularis*.

## 4. Materials and Methods

### 4.1. Plant Material

A total of 552 plants from 69 populations and 28 weed species were used in this study. Eighteen wild and mutant populations from three species were employed for the initial development of the dPACS assays (Table 1). Subsequently, 24 species from Syngenta’s in-house screening collection, each represented by a single wild-type herbicide-sensitive population, were utilised for testing the transferability and wide applicability of the newly developed method. The species employed in the transferability study consisted of 12 grass (*Apera spica-venti*, *Avena fatua*, *Echinochloa crus-galli*, *Eleusine indica*, *Alopecurus myosuroides*, *Poa annua*, *Setaria viridis*, *Phalaris minor*, *Sorghum halepense*, *Digitaria sanguinalis*, *Bromus tectorum* and *Lolium rigidum*) and 12 broad-leaf (*Conyza bonariensis, Amaranthus palmeri, Ambrosia artemisiifolia, Kochia scoparia, Chenopodium album, Raphanus raphanistrum, Euphorbia heterophylla, Bidens pilosa, Galium aparine, Stellaria media, Polygonum convolvulu,* and *Papaver rhoeas*) weeds. Finally, 27 herbicide-resistant populations from six weed species were genotyped with the relevant dPACS assays (Table 2). Seeds from the different populations were sown directly in pots containing a soil medium of compost and peat in a 1:1 ratio. The pots were watered as required and maintained in two different light and temperature-controlled glasshouses depending on their zones of adaptation. Plants acclimatised to warm conditions were grown in a glasshouse bay characterised by a 17 H photoperiod of 180 µmol m^−2^ s^−1^ with temperatures of 25 °C day and 19 °C night and 65% relative humidity. The conditions for cold-adapted species were as follows: 16 H photoperiod of 180 µmol m^−2^ s^−1^ with temperatures of 20 °C day and 16 °C night and 65% relative humidity.

### 4.2. Establishment of Derived Polymorphic Amplified Cleaved Sequence (dPACS) Assays

Two dPACS assays were developed for detecting the nucleotides that code for wild-type herbicide-sensitive amino acid residues S264-*psb*A in *Amaranthus retroflexus* and I2041-ACCase in *Lolium multiflorum*. A third dPACS assay was aimed at detecting wild type herbicide sensitive P106-EPSPS and its S/T/A106 glyphosate resistance-causing variants in *E. indica.* S264-*psb*A is encoded by the AGT triplet in all plant species. The most commonly encountered target-site-resistance mutation is serine 264 to glycine change in the corresponding D1 protein of photosystem II as a consequence of an adenine to guanine substitution at the first base of the codon triplet [33]. The codon triplet for I2041 is generally ATT (and sometimes ATC; e.g., GenBank reference: AJ966445) in grass weeds which is most often changed into AAT resulting in a mutant asparagine amino acid residue [79]. Wild type P106-EPSPS and the S106, T106 and A106-EPSPS variants are coded by the CCN, TCN, ACN and GCN triplets respectively [82,83]. Eight plants per six diverse populations for each species were used to develop the different dPACS assays (Table 1).

#### 4.2.1. DNA Extraction

A 1-cm leaf segment from individual plants was collected in a 96 deep-well assay block (Costar, Cambridge, MA, USA), frozen to −80 °C and then used to extract DNA for downstream PCR amplification. Stainless steel beads (Qiagen, Manchester, UK) were added to each well before the plant material was ground on a Spex Certiprep 2010 model Geno/Grinder (Metuchen, NJ, USA). DNA from the ground material was subsequently extracted using a KingFisher^TM^ Flex Purification system (ThermoFisher Scientific, Waltham, MA, USA) and reagents from a Wizard Magnetic 96 DNA Plant System kit (Promega, Madison, WI, USA).

#### 4.2.2. Nucleotide Sequences Around the SNPs of Interest

The nucleotide variability around the 264-*psb*A, 2041-ACCase and 106-EPSPS codons was determined for *A. retroflexus*, *L. multiflorum* and *E. indica* respectively in order to design robust and widely applicable PCR primers across different populations and plants and to validate the results generated with the dPACS assays. Eight individual plants from each of the six diverse populations were analysed (Table 1). The PCR primers used for amplifying a DNA fragment around the three different codons are summarised in Table 3. PCR was carried out on the extracted DNA using PuReTaq Ready-To-Go PCR beads (Amersham Biosciences, Bucks, UK) in a total volume of 25 µL containing 0.8 µM of each primer and about 50 ng of genomic DNA. The reactions were performed on a Master Cycle Gradient Thermocycler Model 96 (Eppendorf, Stevenage, UK) set with a denaturation step at 95 °C for 4 min followed by 30 cycles of 30 s at 95 °C, 30 s at 60 °C and 1 min at 72 °C. A final extension step for 10 min at 72 °C was also included. Direct Sanger sequencing (Genewiz LLC, South Plainfield, NJ, USA) was subsequently carried out on 1 μL of neat PCR product using the forward PCR primer. The 48 individual sequences (8 plants × 6 populations) per gene fragment were aligned and compared using the Seqman software (version, DNASTAR Lasergene 10, DNASTAR, Madison, WI, USA).

#### 4.2.3. Development of the dPACS 1.0 Software and Selection of Distinguishing Primer/Enzyme Combinations

A simple program (dPACS 1.0) was written to facilitate the design of PCR primers and the selection of corresponding restriction enzymes for use in the dPACS methodology. The program is essentially a perl script wrapper around the EMBOSS (http://emboss.sourceforge.net/) programs fuzznuc and restrict. A copy of the REBASE (http://rebase.neb.com/rebase/rebase.files.html) database is reformatted at runtime into several files suitable for fuzznuc and used to scan the wildtype and mutant sequences for differences. The restrict program is used in certain edge cases involving sequences with gaps. The script is designed to run from the command line, within Galaxy (https://galaxyproject.org/) or as a perl cgi. The dPACS 1.0 program is freely accessible via a web-server (http://opendata.syngenta.agroknow.com/models/dpacs) and the source code is deposited in Syngenta’s public GitHub repository (https://github.com/syngenta). Wild and mutant sequences differing at a single or few concurrent nucleotides are submitted in the dPACS 1.0 program as well as the number of desired mismatches on the forward, reverse or both primers. Indels can also be handled and are entered as hyphens in the place of nucleotides. The program then generates a number of primer and enzyme combinations from which the experimenter can choose from, taking into consideration the robustness of the resulting assay and the commercial availability, cost and strength of the restriction enzymes. In particular, the number of forced mutations are kept to a minimum and as far as possible away from the 3′ end of the primers to ensure good level of targeted PCR amplification. In this current study, the primers were designed so that the wild type amplicons are restricted for the S264-*psb*A, I2041-ACCase and P106-EPSPS assays. Additional nucleotide mismatches were incorporated on the forward EPSPS primer as necessary to allow for the positive identification of serine106, threonine106 and alanine106 variants that endow resistance to glyphosate. The different EPSPS assays use a single PCR product but four different restriction enzymes to differentiate between proline wild type and the serine, threonine and alanine variants. The primers and corresponding enzymes employed are listed in Table 3. The different dPACS assays developed here are schematically shown on Figure 1.

#### 4.2.4. Three-Step PCR, Restriction Digestion and Gel Electrophoresis

PCR was carried out on DNA extracted from eight plants per population using PuReTaq Ready-To-Go PCR beads (Amersham Biosciences, Bucks, UK) in a total volume of 25 µL containing 0.8 µM of each primer and about 50 ng of genomic DNA. The reactions were performed on a Master Cycle Gradient Thermocycler Model 96 (Eppendorf, UK). A hot start was applied with a denaturation at 95 °C for 4 min followed by 30 cycles of 30 s at 95 °C, 30 s at 60 °C and 1 min at 72 °C. A final extension step for 10 min at 72 °C was also included. Three microlitres of neat PCR product was digested with 7.5 units of each restriction enzyme (New England Biolabs, Hertfordshire, UK) in a total volume of 20 µl according to the manufacturer’s recommendations and analysed on 4% MetaPhor^TM^ agarose gel (Lonza, Walkersville, MD, USA) containing 0.5 µg mL^−1^ ethidium bromide run for 1 h at 80 V with 1× TBE buffer.

### 4.3. Transferability of the dPACS Assays to a Wide Range of Weed Species

The dPACS assays initially designed on nucleotide sequences from a single weed species were tested on a diverse set of relevant species. For each species eight individual plants from a single herbicide-sensitive population were tested. The S264-*psb*A and P/S/T/A106-EPSPS assays were tried on 12 grass and 12 broad-leaf species using the same DNA extraction, PCR-RFLP and gel electrophoresis conditions as with *A. retroflexus* and *E. indica* respectively. Following low intensity of PCR amplification regarding the EPSPS assay with broad-leaf weeds using the *E. indica* primers, a new set of *Amaranthus tuberculatus*-based PCR primers (Forward: 5′ GATTCAACTTTTCCTTGGAAATGCAGGAACAGGAATGCGT3′; reverse: 5′GCTTGAATTTCCTCCAGCAACGGCAACCCAAGCTGTCAAT3′) were designed and tested on all the other dicotyledonous weeds. The 2041-ACCase assay was attempted on the 12 grass weeds only as dicot species are characterised by a significantly different enzyme which is not sensitive to ACCase-inhibiting herbicides at the rates applied in the field [108].

### 4.4. Genotyping Weed Populations Resistant to PSII, ACCase and EPSPS-Inhibiting Herbicides

Twenty-seven herbicide resistant populations from six different species were genotyped with the three established dPACS assays to determine whether a mutation at the tested codon position was responsible for or was a contributing factor in herbicide failure in the different samples (Table 2). The 264-*psb*A assay was used to characterise one *C. album* and two *Solanum nigrum* populations with known resistance to the photosystem II herbicide ametryn. The 2041-ACCase dPACS assay was employed to test six each of *S. viridis* and *E. indica* populations which are all resistant to the ACCase-inhibiting herbicide haloxyfop-p-methyl. The wild type P106-EPSPS and, as necessary, the mutant S106-EPSPS, T106-EPSPS and A106-EPSPS assays were used to genotype six each of *Digitaria insularis* and *A. palmeri* populations which are resistant to the herbicide glyphosate. In all cases, eight plants were tested per population.

## 5. Conclusions

We have developed a novel PCR-RFLP procedure and a corresponding computer program for genotyping known SNPs and DIPs. The dPACS technique offers a number of advantages over the CAPS/dCAPS methodologies, including a wider range of primer/enzyme combinations for differentiating wild from mutant DNA sequences. The assays designed on one species were highly transferable to other weeds, thereby circumventing the need for target gene sequence information for identifying the same mutations in a new species. The assays established here will form part of a growing arsenal of molecular tools for investigating the mechanism and the monitoring of resistance to commonly employed EPSPS, Photosystem II and ACCase-inhibiting herbicides [109]. In particular, the P106X assays will be very useful for detecting early signs of glyphosate resistance in the field. This is because mutant individuals, especially from polyploid species, may not always survive a glyphosate application but when combined with other resistant traits or challenged under sub-optimal spray conditions, will escape the herbicide treatment [81,110]. The simple and cost-effective dPACS approach will also serve many other disciplines that analyse SNPs and DIPs.

## Figures and Tables

**Figure 1 ijms-20-03193-f001:**
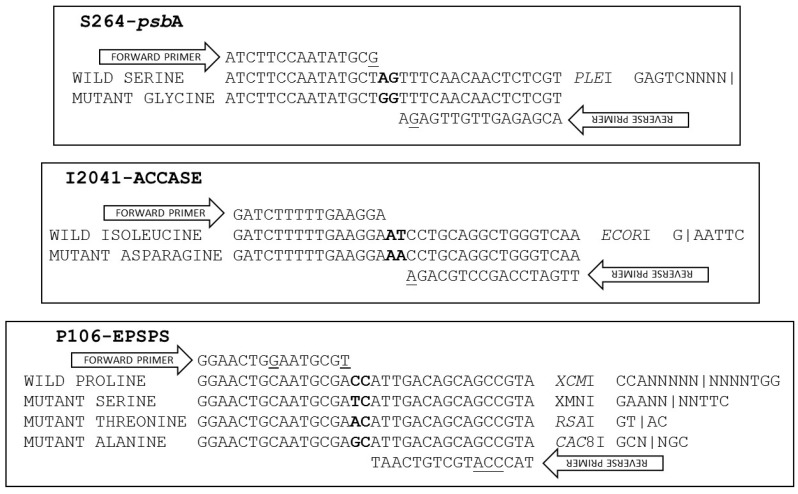
Schematic representation of the dPACS assays developed in this study. The targeted diagnostic di-nucleotides on the wild and mutant DNA sequences are in bold. Forces mismatches on the primers are underlined.

**Figure 2 ijms-20-03193-f002:**
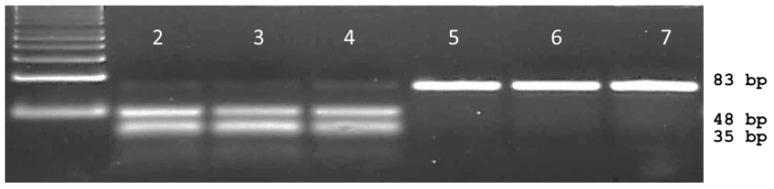
Typical dPACS profiles for wild S264 and mutant G264 *psb*A *Amaranthus retroflexus* individuals. Lane 1: 50 bp DNA ladder, lanes 2, 3, 4 *Ple*I restricted wild S264 haplotype; lanes 5, 6 and 7 *Ple*I unrestricted mutant G264 haplotype.

**Figure 3 ijms-20-03193-f003:**
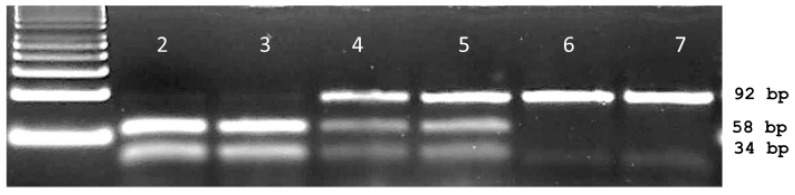
Typical dPACS profiles for wild and I2041N mutant ACCase *Lolium multiflorum* individuals. Lane 1: 50 bp DNA ladder, lanes 2 and 3 homozygous wild II2041 *Eco*RI restricted PCR fragment; lanes 6 and 7 homozygous mutant NN2041 unrestricted PCR fragment; lanes 4 and 5 heterozygous IN2041 plants containing one each of the unrestricted and restricted fragment.

**Figure 4 ijms-20-03193-f004:**
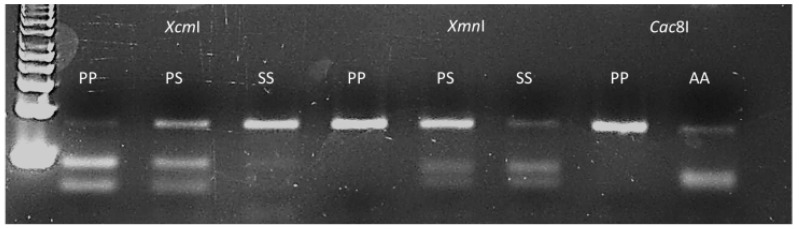
Typical dPACS profiles for wild and P106X mutant EPSPS *Eleusine indica* individuals. Lane 1: 50 bp DNA ladder, lanes 2, 3 and 4 homozygous wild PP106, heterozygous mutant PS106 and homozygous mutant SS106 individuals respectively as revealed by discriminative enzyme *Xcm*I; lanes 5, 6 and 7 homozygous wild PP106, heterozygous mutant PS106 and homozygous mutant SS106 individuals respectively as revealed by discriminative enzyme *Xmn*I; lanes 8 and 9 homozygous wild PP106 and homozygous mutant AA106 individuals respectively as revealed by discriminative enzyme *Cac*8I.

**Figure 5 ijms-20-03193-f005:**
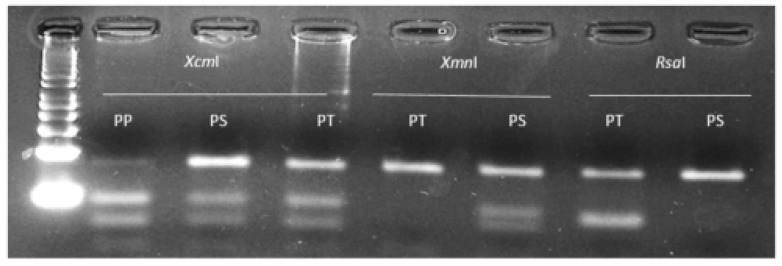
Typical dPACS profiles for wild and P106X mutant EPSPS *Digitaria insularis* individuals. Lane 1: 50 bp DNA ladder, lanes 2, 3 and 4 homozygous wild PP106, heterozygous mutant PS106 and heterozygous mutant PT106 individuals respectively as revealed by discriminative enzyme *Xcm*I; lanes 5, 6 heterozygous mutant PT106 and heterozygous mutant PS106 individuals respectively as revealed by discriminative enzyme *Xmn*I; lanes 7 and 8 heterozygous mutant PT106 and heterozygous mutant PS106 individuals respectively as revealed by discriminative enzyme *Rsa*I.

**Table 1 ijms-20-03193-t001:** Species and populations used to validate the three derived Polymorphic Amplified Cleaved Sequence’ (dPACS) assays and corresponding Sanger sequencing and dPACS results.

Species	Population	Origin	dPACS and Sanger Sequencing Assays	Genotypic Frequencies as Revealed by Sequencing and dPACS Assays
*Amaranthus retroflexus*	ArD1	USA	S264G *psb*A	100% mutant G264
	ArD2	Switzerland	S264G *psb*A	100% wild S264
	ArD3	USA	S264G *psb*A	100% wild S264
	ArD4	Switzerland	S264G *psb*A	100% mutant G264
	ArD5	Switzerland	S264G *psb*A	100% wild S264
	ArD6	Switzerland	S264G *psb*A	100% mutant G264
*Lolium multiflorum*	LmD1	UK	I2041N ACCase	25%II2041, 50%IN2041, 25%NN2041
	LmD2	UK	I2041N ACCase	100% II2041
	LmD3	UK	I2041N ACCase	100% II2041
	LmD4	UK	I2041N ACCase	100% II2041
	LmD5	UK	I2041N ACCase	100% NN2041
	LmD6	UK	I2041N ACCase	12.5% II2041, 87.5% NN2041
*Eleusine indica*	EiD1	Philippines	P106S/T/A EPSPS	100% SS106
	EiD2	Malaysia	P106S/T/A EPSPS	100% SS106
	EiD3	Ecuador	P106S/T/A EPSPS	100 % AA106
	EiD4	Malaysia	P106S/T/A EPSPS	100 % PP106
	EiD5	Malaysia	P106S/T/A EPSPS	50% PP106, 25% PS106, 25% SS106
	EiD6	Malaysia	P106S/T/A EPSPS	25% PP106, 75% SS106

**Table 2 ijms-20-03193-t002:** Genotypic frequencies of wild and mutant *psb*A, acetyl-CoA carboxylase (ACCase) and 5-enolpyruvylshikimate-3-phosphate synthase (EPSPS) alleles as determined by corresponding dPACS assays in 27 herbicide-resistant populations from six different species.

Species	Population	Origin	dPACS Assay	Genotypic Frequencies
*Chenopodium album*	CaR1	Poland	S264G *psb*A	100% G264
*Solanum nigrum*	SnR1	Germany	S264G *psb*A	100% G264
	SnR2	Unknown	S264G *psb*A	100% G264
*Setaria viridis*	SvR1	USA	I2041N ACCase	100% II2041
	SvR2	USA	I2041N ACCase	87.5% II2041, 12.5% IN2041
	SvR3	Canada	I2041N ACCase	100% II2041
	SvR4	USA	I2041N ACCase	100% II2041
	SvR5	Canada	I2041N ACCase	100% II2041
	SvR6	USA	I2041N ACCase	100% II2041
*Eleusine indica*	EiR1	Malaysia	I2041N ACCase	100% II2041
	EiR2	Philippines	I2041N ACCase	87.5% II2041, 12.5 % NN2041
	E2R3	Malaysia	I2041N ACCase	87.5% II2041, 12.5 % NN2041
	EiR4	Ecuador	I2041N ACCase	100% II2041
	EiR5	Malaysia	I2041N ACCase	100% II2041
	EiR6	Malaysia	I2041N ACCase	75% II2041, 12.5% IN2041, 12.5% NN2041
*Amaranthus palmeri*	ApR1	USA	P106S/T/A EPSPS	100% PP106
	ApR2	USA	P106S/T/A EPSPS	100% PP106
	ApR3	USA	P106S/T/A EPSPS	100% PP106
	ApR4	USA	P106S/T/A EPSPS	100% PP106
	ApR5	USA	P106S/T/A EPSPS	100% PP106
	ApR6	USA	P106S/T/A EPSPS	100% PP106
*Digitaria insularis*	DiR1	Brazil	P106S/T/A EPSPS	100% PS106
	DiR2	Brazil	P106S/T/A EPSPS	100% PS106
	DiR3	Brazil	P106S/T/A EPSPS	100% PS106
	DiR4	Brazil	P106S/T/A EPSPS	100% PT106
	DiR5	Brazil	P106S/T/A EPSPS	100% PS106
	DiR6	Brazil	P106S/T/A EPSPS	100% PS106

**Table 3 ijms-20-03193-t003:** Sequencing and dPACS primers employed and corresponding PCR products and restriction digests. The forced nucleotide mismatches on the primers are in bold and underlined.

Primer ID		Primer Length (bp)	Primer Sequence 5’–3’	Target Codon and Amino	PCR Product, Restriction Enzyme and Digest
Sequencing primers	SeqFw-*psb*A-*A.retroflexus*	25	AAGGAAGCTTTTCTGATGGTATGCC	264 *psb*A	426 bp; codons 165–307
	SeqRv-*psb*A-A.retroflexus	25	CTACAGATTGGTTGAAGTTGAAACC		
	SeqFw-ACCase-*L. multiflorum*	22	TGGCAGAGCAAAACTTGGAGGG	2041 ACCase	468 bp; codons 1954–2110
	SeqRv-ACCase-*L. multiflorum*	23	CTGAACTTGATCTCAATCAACCC		
	SeqFw-EPSPS-*E. indica*	22	GGAACAACTGTGGTGGATAACC	106 EPSPS	492 or 498 bp; codons 39–171
	SeqRv-EPSPS-*E. indica*	20	CTTGCCACCAGGTAGCCCTC		
dPACS primers	Fw-PsbA-*A.retroflexus*	40	TGCTTCATGGTTACTTTGGTCGATTGATCTTCCAATATGCG	S264 *psb*A	*Ple*I (GAGTCN4|)
	Rv-PsbA-*A.retroflexus*	41	AAGCAGCTAAGAAAAAGTGTAAAGAACGAGAGTTGTTGAGA		83 bp undigested; 48/35 bp digested
	Fw-ACCase-*L. multiflorum*	35	GCTTCTCTGGTGGGCAAAGAGACCTTTTTGAAGGA	I2041 ACCase	*Ecor*I (G|AATTC)
	Rv-ACCase-*L. multiflorum*	55	GGCAGGCAGATTATATGTCCTAAGGTTCTCAACAATTGTTGATCCAGCCTGCAGA		92 bp undigested; 58/34 bp digested
	Fw-EPSPS-E. indica	40	GGTGCAGCTCTTCTTGGGGAATGCTGGAACTGGAATGCGT	P106 EPSPS	*Xcm*I (CCA(NNNNN|NNNN)TGG
	Fw-EPSPS-*E. indica*	40	AGTTGCATTTCCTCCAGCAGCAGTTACCCATGCTGTCAAT		82 bp undigested; 47/35 bp digested
	Rv-EPSPS-*E. indica*	40	GGTGCAGCTCTTCTTGGGGAATGCTGGAACTGGAATGCGT	S106 EPSPS	*Xmn*I (GAANN|NNTTC)
	Rv-EPSPS-*E. indica*	40	AGTTGCATTTCCTCCAGCAGCAGTTACCCATGCTGTCAAT		82 bp undigested; 45/37 bp digested
	Fw-EPSPS-*E. indica*	40	GGTGCAGCTCTTCTTGGGGAATGCTGGAACTGGAATGCGT	T106 EPSPS	*Rsa*I (GT|AC)
	Rv-EPSPS-*E. indica*	40	AGTTGCATTTCCTCCAGCAGCAGTTACCCATGCTGTCAAT		82 bp undigested; 42/40 bp digested
	Fw-EPSPS-*E. indica*	40	GGTGCAGCTCTTCTTGGGGAATGCTGGAACTGGAATGCGT	A106 EPSPS	*Cac8*I (GCN|NGC)
	Rv-EPSPS-*E. indica*	40	AGTTGCATTTCCTCCAGCAGCAGTTACCCATGCTGTCAAT		82 bp undigested; 43/39 bp digested

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
