# Peer review of "Derived Polymorphic Amplified Cleaved Sequence (dPACS): A Novel PCR-RFLP Procedure for Detecting Known Single Nucleotide and Deletion–Insertion Polymorphisms"

_ijms, 2019, doi:10.3390/ijms20133193_

Reviewer 1 Report

Sentence starting line 44 in the introduction asserting that intergenic regions are phenotypically silent is too strong and borders on inaccurate. Many GWAS identified SNPs associated with disease risks are in intergenic regions. Moreover, both the intracellular organisation of packed chromatin and the high extent of transcription from across the genome, the latter leading to non coding RNA leave plenty of room for functional consequences outside of regions classically defined as genes.

Sequencing was performed originally across the genes of interest to identify variants for subsequent assay development. One feature of the method presented is its potential to be robust even in the presence of variants occurring elsewhere corresponding to regions of the primers outside of the locations for polymorphisms of interest. I would have found it interesting if this had been explored further, for example by sequencing the corresponding regions in a selected number of the test samples to determine how many bases, if any, were variant in the body of the primers. Sequencing the resulting PCR products of course only gives rise to the original primer sequences plus targeted variants of interest. Even in the absence of further data, this could have been more obviously discussed.

Existing methods are discussed and compared in detail in relation to the new method. There is a lot of detail, the paper felt very long, it may that it could be reduced in size by restructuring / greater use of figures / tables

Author Response

Reviewers comment 1:

Sentence starting line 44 in the introduction asserting that intergenic regions are phenotypically silent is too strong and borders on inaccurate. Many GWAS identified SNPs associated with disease risks are in intergenic regions. Moreover, both the intracellular organisation of packed chromatin and the high extent of transcription from across the genome, the latter leading to non coding RNA leave plenty of room for functional consequences outside of regions classically defined as genes.

Response to reviewers comment 1:

The sentence has been softened into: SNPs that occur in inter-genic regions are generally silent phenotypically ....

Reveiwers comment 2:

Sequencing was performed originally across the genes of interest to identify variants for subsequent assay development. One feature of the method presented is its potential to be robust even in the presence of variants occurring elsewhere corresponding to regions of the primers outside of the locations for polymorphisms of interest. I would have found it interesting if this had been explored further, for example by sequencing the corresponding regions in a selected number of the test samples to determine how many bases, if any, were variant in the body of the primers. Sequencing the resulting PCR products of course only gives rise to the original primer sequences plus targeted variants of interest. Even in the absence of further data, this could have been more obviously discussed.

Response to reviewers comment 2:

As indicated in the text (section 4.2.2 line 501 and table 3 sequencing was not carried out on fragments that were generated using dPACS primers. Instead, more than 400 bp fragments were generated  for each of the 3 trageted genes with sequencing primers. The fragments comprised of at least 100 bp on each side of the SNP of interest to allow the gathering of nucleotide sequence information for subsequent robust primer design. Also as indicated in the text on line 488, in spite of as many as five mismtaches across the two forward and reverse EPSPS primers, a strong amplification was observed for the dPACS PCR fragment

Reveiwers comment 3:

Existing methods are discussed and compared in detail in relation to the new method. There is a lot of detail, the paper felt very long, it may that it could be reduced in size.

Response to reviewers comment 3:

The dPACS method is widely applicable across the many disciplines that require the analyses of SNPs and DIPs. The discussion may appear a bit long but the authors thought that it was important to clearly exemplify and justify the many instances where the novel dPACS method could be applied .

 Reviewer 2 Report

Kaundun and coauthors describe and use a method (dPACs) for a low/medium throughput  genotyping (PCR-RFLP based genotyping) that is very versatile (in terms of solution space and related costs). The authors also show application examples.

The paper is clear and well described.

Minor comment

The section of the discussion is very rich but I find it too long and does not make reading easy or fluent. I would suggest to shorten it, while keeping alive all the application examples and the advantages compared to current methods.

Major comment

1 - Primer construction - base modified at the 3 'end with respect to the reference.

I was surprised that the method involves primers with modified bases (Figure 1: S264-psbA forward primer; I2041- ACCASE reverse primer; P106-EPSPS forward primer) at the 3' end and that this still allows the amplification of the fragment. I imagine that in these cases, less stringent amplification conditions should be used (if so). I have not  seen this important issue to be discussed and I believe that a brief discussion/comment on this point  would make readers more confident on the method.

Author Response

Reviewers comment 1:

The section of the discussion is very rich but I find it too long and does not make reading easy or fluent. I would suggest to shorten it, while keeping alive all the application examples and the advantages compared to current methods.

Response to comment 1:

The dPACS method is widely applicable across the many disciplines that require the analyses of SNPs and DIPs. The discussion may appear a bit long but the authors thought that it was important to clearly exemplify and justify the many instances where the novel dPACS method could be applied .

Reviewers comment 2:

I was surprised that the method involves primers with modified bases (Figure 1: S264-psbA forward primer; I2041- ACCASE reverse primer; P106-EPSPS forward primer) at the 3' end and that this still allows the amplification of the fragment. I imagine that in these cases, less stringent amplification conditions should be used (if so). I have not  seen this important issue to be discussed and I believe that a brief discussion/comment on this point  would make readers more confident on the method.

Response to comment 2:

Mismatches on 3' end of primers do always result in PCR failure if carefully chosen as is the case of this study. Such mismatches are also introduced for the published dCAPS method (Neff, M. M.; Neff, J. D.; Chory, J.; Pepper, A. E., dCAPS, a simple technique for the genetic analysis of single nucleotide polymorphisms: Experimental applications in Arabidposis thaliana genetics. Plant J 1998, 14, (3), 387-392.). As explained in the text, the novel dPACS method results in strong PCR amplification because short 70-120 bp fragments are targeted and also due to the fact that the primers collectively cover the entire length of the targeted sequence except for 1 or 2 base pairs being interrogated (lines 87-89).